# Role of thyroid autoimmunity independent of thyroid hormone levels in pulmonary function decline: A population-based study in euthyroid adults

**Kyung-Hun Sung**[1], **Jeongeun Kwak**[2], **Mee Kyung Kim**[2], **Dong-Jun Lim**[1], **Jung-Min Lee**[2], **Sang-Ah Chang**[2], **Jeongmin Lee**[2]*

**1** Division of Endocrinology and Metabolism, Department of Internal Medicine, Seoul St. Mary's Hospital, College of Medicine, The Catholic University of Korea, Seoul, Republic of Korea, **2** Division of Endocrinology and Metabolism, Department of Internal Medicine, Eunpyeong St. Mary's Hospital, College of Medicine, The Catholic University of Korea, Seoul, Republic of Korea

\* 082mdk45@catholic.ac.kr

## Abstract

### Purpose

Although overt thyroid dysfunction has been associated with changes in pulmonary function, the effects of thyroid hormone levels and thyroid autoimmunity on lung function in euthyroid individuals remain unclear. We investigated the associations between subtle changes in thyroid hormones and thyroid peroxidase antibodies (TPOAb) and pulmonary function in a nationally representative cohort of Korean adults.

### Methods

We analyzed data from 2,626 euthyroid participants aged ≥ 40 years from the Korea National Health and Nutrition Examination Survey (2013–2015). Pulmonary function was assessed using spirometry-derived forced vital capacity (FVC), forced expiratory volume in 1 second (FEV1), and FEV1/FVC ratio. Multivariable linear regression analyses were conducted after adjusting for age, sex, body mass index (BMI), smoking status, alcohol intake, and physical activity.

### Results

In the unadjusted models, higher free thyroxine levels were positively associated with FVC and FEV1, while higher TSH levels were inversely associated. In contrast, elevated TPOAb levels were independently associated with reduced FEV1 ($\beta = -0.330$, $P = 0.021$). These associations were more pronounced among adults aged ≥60 years and individuals with a BMI ≤ 23 kg/m².

**Data availability statement:** The data used in this study were obtained from the sixth Korea National Health and Nutrition Examination Survey (KNHANES, 2013–2015), which is owned and managed by the Korea Disease Control and Prevention Agency (KDCA). The anonymized KNHANES dataset is publicly available, and any qualified researcher can download the data from the official KDCA website. Qualified researchers can register and apply for access through this portal following the instructions outlined on the site. The website includes detailed manuals and guidelines for data use, although these materials are currently in Korean. (https://knhanes.kdca.go.kr).

**Funding:** This work was supported by the National Research Foundation of Korea (NRF) grant funded by the Korea government (MSIT) (RS-2025-16067443) for Jeongmin Lee. The funders had no role in study design, data collection and analysis, decision to publish, or preparation of the manuscript.

**Competing interests:** The authors have declared that no competing interests exist.

## Conclusion

Thyroid autoimmunity, as reflected by elevated TPOAb levels, was associated with a decline in pulmonary function among euthyroid individuals, independent of thyroid hormone levels. Our results support the clinical utility of TPOAb status as an early marker for detecting subclinical pulmonary vulnerability, particularly in older adults and those with a lower BMI.

## Introduction

The thyroid gland primarily secretes thyroxine (T4) and biologically active triiodothyronine (T3), which exert direct cellular effects by binding to nuclear receptors [1]. Thyroid hormones are crucial regulators of multiple organ systems, including metabolic activity, cardiovascular function, neuromuscular control, thermoregulation, and oxygen consumption [2]. Given the close interaction between oxygen metabolism and pulmonary function, it is plausible that thyroid hormones influence respiratory physiology. In hypothyroidism, decreased levels of T3 and T4 have been shown to reduce respiratory drive and impair the strength and coordination of respiratory muscles, leading to decreased pulmonary function, particularly forced vital capacity (FVC) and forced expiratory volume in one second (FEV1), which is consistent with a restrictive ventilatory pattern [3,4]. Conversely, hyperthyroidism is often associated with increased oxygen consumption, respiratory alkalosis, and sometimes, a heightened ventilatory response [5].

Anti-thyroid peroxidase antibodies (TPOAb) are the most prevalent markers of autoimmune thyroiditis and are detectable in a substantial proportion of euthyroid individuals [6]. Accumulating evidence suggests that TPOAb is involved in systemic immunological effects beyond the thyroid gland. A previous study has reported an association between thyroid antibodies and interstitial lung disease. Increased levels of circulating cytokines such as interleukin-6 and tumor necrosis factor- α have been observed in association with thyroid autoimmunity. The cross-reactivity between thyroid antibodies and pulmonary antigens has also been suggested as a potential mechanism [7]. Decreased lung function has also been reported in euthyroid individuals with thyroid antibody positivity, suggesting that subclinical immune dysregulation may contribute to respiratory dysfunction [8].

Despite these observations, prior studies have often been limited by small sample sizes or insufficient adjustment for confounding variables [4,9–12]. To address these limitations, this study used nationally representative data to examine the association between thyroid hormones, TPOAb, and pulmonary function in individuals without diagnosed thyroid disorders. By controlling for key lifestyle factors, we aimed to clarify the broader physiological relationship between thyroid and pulmonary function.

## Materials and methods

### Study population and data source

This retrospective study used data from the sixth Korea National Health and Nutrition Examination Survey (2013–2015), a nationwide cross-sectional survey employing



stratified, multistage probability sampling. The KNHANES includes a health interview, health examination, and nutrition survey, offering comprehensive data on the health status, behaviors, and nutrition of the Korean population [13]. In our study, the initial cohort included 2,895 individuals who underwent health examinations during the KNHANES VI period. Participants were excluded if they had a diagnosis of thyroid disease (n = 158), abnormal free T4 (FT4) levels (n = 83), a previous prescription for asthma medication (n = 46), or treatment for chronic obstructive pulmonary disease (COPD) (n = 9). Ultimately, the final study population included 2,626 subjects (Fig 1). Data were accessed for research purposes between April 11 and Aprii 13, 2025. This study complied with the ethical standards of the Declaration of Helsinki and was approved by the Catholic University of Korea, Catholic Medical Center, Eunpyeong St. Mary's Hospital Institutional Review Board (approval no. PC25ZISI0065, April 10th 2025). Informed consent was waived because the analyses were conducted using anonymized data.

## Laboratory analyses

All blood samples were collected in the morning after at least 8 hours of fasting, promptly processed, centrifuged, aliquoted, and transported to the Central Testing Institute in Seoul for biochemical analysis within 24 hours. Thyroid function tests were conducted using electrochemiluminescence immunoassay kits from Roche Diagnostics (Basel, Switzerland): E-TSH (RRID: AB_2756377), E-Free T4 (RRID: AB_2801661), and E-Anti-TPO (RRID: AB_2916057). Fasting spot urine samples, mainly first-morning midstream specimens, were analyzed for iodine status. Urinary iodine concentration (UIC) was measured using inductively coupled plasma mass spectrometry (PerkinElmer, Waltham, MA, USA) with standardized iodine reference material (Inorganic Venture, Christiansburg, VA, USA), and adjusted to the iodine-to-creatinine ratio (UICR, μg/g).

The standard reference ranges for thyroid function, based on assay kit specifications, were 0.81–1.76 ng/mL for free T4 and 0.35–5.50 mIU/L for TSH. However, due to Korea's high iodine intake levels, the TSH reference range was adjusted to 0.62–6.86 mIU/L, corresponding to the 2.5th–97.5th percentiles derived from the KNHANES VI population [14]. The

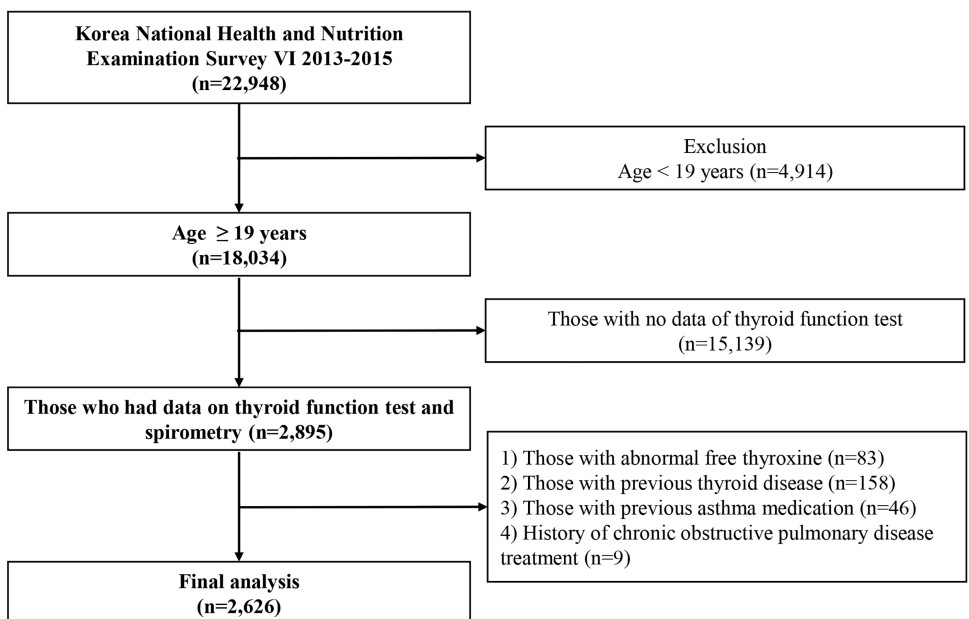

**Fig 1. Flow chart of the study population.**



reference range for TPOAb was 0–34 IU/mL. Internal quality control procedures were conducted monthly to ensure coefficient of variation values remained within acceptable limits [13]. External quality assurance was provided by both the College of American Pathologists and Korean Association of External Quality Assessment Services. Detailed information on laboratory quality control is available on the official KNHANES website (https://knhanes.kdca.go.kr).

## Measurements and definitions

Body mass index (BMI) was calculated as weight (kg) divided by height in meters squared (m²). Smoking status and alcohol use were assessed using self-administered questionnaires. Smoking status was categorized as current, former, or never smoker, according to the definitions established by the Korea Centers for Disease Control and Prevention. Participants were categorized as non-smokers if they had never smoked or had smoked fewer than 100 cigarettes in their lifetime. Former smokers had smoked at least 100 cigarettes but had quit smoking at the time of the survey. Current smokers had smoked at least 100 cigarettes and were still smoking at the time of the data collection. Alcohol consumption was assessed to determine risky drinking behavior, which was defined as an average weekly intake of ≥210 g of alcohol per week for men and ≥140 g of alcohol per week for women.

In the KNHANES before 2014, the International Physical Activity Questionnaire (IPAQ) was used as the physical activity recall questionnaire to measure the amount of physical activity [15]. Participants reported the frequency and duration of vigorous, moderate, and walking activities performed over the past seven days. Each activity type was assigned a metabolic-equivalent (MET) value in accordance with the IPAQ scoring guidelines, as follows: 3.3 METs for walking, 4.0 METs for moderate-intensity activities, and 8.0 METs for vigorous-intensity activities. Weekly energy expenditure via physical activity was calculated in MET-minutes per week by multiplying the assigned MET value by minutes per day and the number of days per week in which the activity was performed. The total physical activity score was obtained by summing the MET-min/week values for all the activity categories. Physical activity levels were categorized into three groups: low, moderate, and high. Participants were classified as having high physical activity if they either (1) engaged in vigorous-intensity activity for at least three days and achieved a total of 1,500 MET-min/week, or (2) participated in a combination of walking, moderate, and vigorous activity for at least seven days, accumulating at least 3,000 MET-min/week. The moderate physical activity group included individuals who met at least one of the following criteria: (1) vigorous activity for at least 20 min/day on at least three days/week; (2) moderate activity or walking for at least 30 min/day on at least five days/week; or (3) any combination of walking, moderate, or vigorous activity totaling at least 600 MET-min/week. Participants who did not meet the criteria for moderate or high activity were classified as having low physical activity [16].

## Statistical analysis

Statistical analyses were conducted in accordance with the complex multistage sampling design of KNHANES, incorporating sampling weights to generate nationally representative estimates. The SAS® PROC SURVEY procedure was employed to account for stratification, clustering, and unequal probability weighting inherent in the survey design. Demographic and lifestyle variables were analyzed using a complex sample-adjusted Pearson's chi-square ($\chi^2$) test. Multivariable linear regression models were used to assess the association between thyroid function and pulmonary function parameters. Regression coefficients (β) with corresponding 95% confidence intervals (CIs) were calculated to evaluate associations. Multivariable logistic regression analyses were conducted using three hierarchical models: Model 1 was adjusted for age and sex; Model 2 included additional adjustments for BMI, smoking status, and alcohol consumption; and Model 3 incorporated urinary iodine concentration and comorbidities. To determine the clinical significance of the analysis, the population was categorized into the following groups: normal and overweight or obese (BMI > 23 kg/m2), and an age range of 40–60 or > 60 years. A two-sided P value < 0.05 was considered statistically significant.



## Results

### Baseline characteristics of the study participants

The characteristics of the study population are summarized in Table 1. The mean age was 54.37 years, with 53.12% of participants were men. Regarding smoking status, 27.14% were current smokers, 24.09% were former smokers, and 48.77% had never smoked. Most individuals (87.72%) were classified as having low physical activity, while 6.02% and 6.26% reported moderate and vigorous activity, respectively. The mean BMI was 24.26±0.07 kg/m². The mean of FT4 levels was 1.21 ng/dL, and the mean TSH level was 2.66±0.04 µIU/mL. The mean concentration of TPOAb was 43.92±218.43 IU/mL. Pulmonary function tests showed a mean predicted FEV1 of 2.95 L, mean FVC of 3.80 L, and an average FEV1/FVC ratio of 0.78.

### Association between thyroid hormones and pulmonary function

In the univariate linear regression analysis, higher FT4 levels were positively associated with both FVC (β=0.926, 95% CI: 0.460–1.392, $P<0.0001$) and FEV1 (β=0.657, 95% CI: 0.327–0.987, $P<0.0001$). Conversely, TSH levels were inversely associated with FVC (β=−0.048, 95% CI: −0.072 to −0.024, $P<0.0001$) and FEV1 (β=−0.030, 95% CI: −0.046 to −0.015, $P<0.0001$) (Table 2) and showed a positive association with the FEV1/FVC ratio (β=0.002, 95% CI: 0.001–0.003, $P=0.001$). TPOAb levels were significantly negatively associated with FVC (β=−0.953, 95% CI: −1.581 to −0.332, $P=0.003$) and FEV1 (β=−0.810, 95% CI: −1.420 to −0.210, $P=0.007$), and positively associated with the FEV1/FVC ratio (β=0.100, 95% CI: 0.010–0.190, $P=0.028$). After adjusting for potential confounding variables (age, sex, BMI, smoking, alcohol intake, physical activity, and UICR), the associations between thyroid hormones and pulmonary function were no longer statistically significant in the multivariate models, whereas TPOAb levels remained significantly associated with reduced FEV1 up to Model 3 (β=−0.330, 95% CI: −0.620 to −0.040, $P=0.021$).

**Table 1. Demographic and clinical characteristics of total study population.**

|  | Unweighted sample size(n=2,626) | Weighted sample size(n=5,110,505) |
|---|---|---|
| Sex (men, %) | 1390 (52.93) | 2714910 (53.12) |
| Age, years | 54.3 0±8.93 | 54.37±0.16 |
| Smoker, n | 2,547 | 6,070,090 |
| Current smoker, n (%) | 591(23.20) | 1647715(27.14) |
| Former smoker, n (%) | 590(23.16) | 1462234(24.09) |
| Never smoker, n (%) | 1366(53.63) | 2960141(48.77) |
| Physical activity | 2,058 | 4,929,524 |
| Low, n (%) | 1838(89.31) | 4324422(87.72) |
| Moderate, n (%) | 115(5.59) | 296733(6.02) |
| Vigorous, n (%) | 105(5.10) | 308370(6.26) |
| BMI, kg/m² | 24.25±3.11 | 24.2 6±0.07 |
| fT4, ng/dL | 1.21±0.16 | 1.21±0.00 |
| TSH, uIU/mL | 2.67±2.00 | 2.66±0.04 |
| TPOAb, IU/mL | 43.12±217.18 | 43.92±218.43 |
| FEV1, L | 2.86±0.68 | 2.95±0.01 |
| FVC, L | 3.69±0.87 | 3.80±0.02 |
| FEV1/FVC | 0.78±0.07 | 0.78±0.00 |

Data are expressed as the mean±SD, median (25–75%), or n (%).

BMI, body mass index; fT4, free thyroxine; TSH, thyroid-stimulating hormone; TPOAb, thyroid peroxidase antibodies; forced expiratory volume in 1 second; FVC, forced vital capacity.



**Table 2. Multivariate regression analyses of factors associated with pulmonary function test.**

| Pulmonary function | FT4 | | TSH | | TPOAb | |
|---|---|---|---|---|---|---|
| | β (95% CI) | *P*-value | β (95% CI) | *P*-value | β (95% CI) | *P*-value |
| FVC | | | | | | |
| Crude | 0.926 (0.460, 1.392) | <0.001 | −0.048 (−0.072, −0.024) | <0.001 | −0.953 (−1.581, −0.332) | 0.003 |
| Model 1 | −0.067 (−0.237, 0.103) | 0.417 | −0.002 (−0.016, 0.012) | 0.656 | −0.620 (−1.182, −0.058) | 0.031 |
| Model 2 | −0.085 (−0.250, 0.080) | 0.309 | −0.002 (−0.016, 0.012) | 0.734 | −0.210 (−0.865, 0.445) | 0.528 |
| Model 3 | −0.097 (−0.280, 0.086) | 0.298 | −0.004 (−0.018, 0.010) | 0.555 | −0.090 (−0.703, 0.523) | 0.771 |
| FEV1 | | | | | | |
| Crude | 0.657 (0.327, 0.987) | <0.001 | −0.030 (−0.046, −0.015) | <0.001 | −0.810 (−1.420, −0.210) | 0.007 |
| Model 1 | −0.062 (−0.242, 0.118) | 0.347 | 0.002 (−0.007, 0.011) | 0.698 | −0.530 (−1.029, −0.031) | 0.038 |
| Model 2 | −0.073 (−0.205, 0.059) | 0.269 | 0.002 (−0.007, 0.011) | 0.629 | −0.420 (−0.790, −0.050) | 0.026 |
| Model 3 | −0.083 (−0.235, 0.069) | 0.291 | −0.003 (−0.012, 0.006) | 0.540 | −0.330 (−0.620, −0.040) | 0.021 |
| FEV1/FVC | | | | | | |
| Crude | −0.021 (−0.039, −0.003) | 0.223 | 0.002 (0.001, 0.003) | 0.001 | 0.100 (0.010, 0.190) | 0.028 |
| Model 1 | −0.005 (−0.020, 0.010) | 0.546 | 0.001 (−0.001, 0.003) | 0.147 | 0.073 (0.006, 0.140) | 0.034 |
| Model 2 | −0.004 (−0.020, 0.012) | 0.628 | 0.001 (−0.001, 0.003) | 0.166 | 0.028 (−0.035, 0.091) | 0.387 |
| Model 3 | −0.004 (−0.021, 0.013) | 0.666 | 0.000 (−0.002, 0.002) | 0.845 | 0.015 (−0.048, 0.078) | 0.642 |

Model 1: adjustment for age and sex; model 2: model 1 + body mass index, smoking, and drinking; model 3: model 2 + comorbidity and urine iodine.

fT4, free thyroxine; TSH, thyroid-stimulating hormone; TPOAb, thyroid peroxidase antibodies; forced expiratory volume in 1 second; FVC, forced vital capacity.

## Subgroup analyses according to sex, age and BMI

Subgroup analyses stratified by sex, age, and BMI were performed. Among men, higher TPOAb levels were independently associated with decreased pulmonary function, including reduced FVC (β = −0.650, 95% CI: −1.280 to −0.020, *P* = 0.042), FEV1 (β = −0.580, 95% CI: −1.150 to −0.010, *P* = 0.028), and an increased FEV1/FVC ratio (β = 0.070, 95% CI: 0.010 to −0.130, *P* = 0.037). Additionally, TSH showed an inverse association with FVC (β = −0.017, 95% CI: −0.033 to −0.001, *P* = 0.042), while FT4 was not significantly associated with any pulmonary function index. In contrast, none of the thyroid function markers demonstrated statistically significant associations with pulmonary function parameters in the women (Table 3).

In age-stratified analyses, FT4 levels showed significant inverse associations with pulmonary function in the ≥ 60-year-old group. After full adjustment, FT4 was negatively associated with FVC (β = −0.398, 95% CI: −0.749 to −0.047, *P* = 0.026), FEV1 (β = −0.457, 95% CI: −0.740 to −0.174, *P* = 0.002), and the FEV1/FVC ratio (β = −0.048, 95% CI: −0.091 to −0.005, *P* = 0.030). In the 40–59-year-old group, TSH was negatively associated with FVC after adjustment (β = −0.013, 95% CI: −0.021 to −0.005, *P* < 0.001), while no significant associations were observed for FT4 or TPOAb (Table 4).



**Table 3. Sex-specific association between thyroid function and pulmonary function.**

| Pulmonary function | FT4 | | TSH | | TPOAb | |
|---|---|---|---|---|---|---|
| | β (95% CI) | *P*-value | β (95% CI) | *P*-value | β (95% CI) | *P*-value |
| Men (n = 1,390) | | | | | | |
| FVC | | | | | | |
| Crude | 0.240 (−0.038, 0.513) | 0.096 | −0.024 (−0.045, −0.003) | <0.001 | −1.100 (−1.950, −0.250) | 0.012 |
| Adjusted | 0.183 (−0.086, 0.453) | 0.187 | −0.017 (−0.033, −0.001) | 0.042 | −0.650 (−1.280, −0.020) | 0.042 |
| FEV1 | | | | | | |
| Crude | 0.275 (0.048, 0.495) | <0.001 | −0.010 (−0.036, 0.016) | 0.426 | −0.950 (−1.700, −0.200) | 0.014 |
| Adjusted | 0.194 (−0.046, 0.434) | 0.107 | −0.006 (−0.222, 0.010) | 0.468 | −0.580 (−1.150, −0.010) | 0.028 |
| FEV1/FVC | | | | | | |
| Crude | 0.018 (−0.010, −0.046) | 0.210 | 0.004 (0.000, 0.008) | 0.048 | 0.120 (0.010, 0.230) | 0.031 |
| Adjusted | 0.011 (−0.086, 0.452) | 0.430 | 0.002 (−0.001, 0.004) | 0.064 | 0.070 (0.010, 0.130) | 0.037 |
| Women (n = 1,236) | | | | | | |
| FVC | | | | | | |
| Crude | −0.170 (−0.443, 0.109) | 0.248 | 0.011 (−0.018, 0.040) | 0.432 | −0.650 (−1.430, 0.130) | 0.101 |
| Adjusted | −0.113 (−0.332, 0.106) | 0.338 | 0.006 (−0.009, 0.021)) | 0.436 | −0.300 (−1.020, 0.420) | 0.031 |
| FEV1 | | | | | | |
| Crude | −0.130 (−0.38, 0.12) | 0.28 | 0.007 (−0.022, 0.036) | 0.645 | −0.500 (−1.250, 0.250) | 0.195 |
| Adjusted | −0.084 (−0.285, 0.117) | 0.392 | 0.004 (−0.012, 0.020) | 0.601 | −0.250 (−0.980, 0.480) | 0.497 |
| FEV1/FVC | | | | | | |
| Crude | 0.002 (−0.020, 0.024) | 0.825 | 0.002 (−0.003, 0.007) | 0.579 | 0.040 (−0.080, 0.160) | 0.517 |
| Adjusted | 0.000 (−0.022, 0.022) | 0.977 | 0.000 (−0.004, 0.004) | 0.782 | 0.010 (−0.090, 0.110) | 0.811 |

Adjusted for age, sex, body mass index, smoking, drinking, comorbidity and urine iodine.

fT4, free thyroxine; TSH, thyroid-stimulating hormone; TPOAb, thyroid peroxidase antibodies; forced expiratory volume in 1 second; FVC, forced vital capacity.

In participants with a BMI ≤ 23 kg/m², TPOAb was significantly associated with both FVC (β = −0.310, 95% CI: −0.033 to −0.001, *P* = 0.019) and FEV1 (β = −0.280, 95% CI: −0.950 to −0.183, *P* = 0.047), while FT4 and TSH showed no significant associations in this group (Table 5). Conversely, among participants with BMI > 23 kg/m², TSH was inversely associated with FEV$_1$ (β = −0.015, 95% CI: −0.029 to −0.001, *P* = 0.039), whereas no significant associations were observed for FT4 or TPOAb.

## Discussion

In this nationwide representative cohort of individuals without thyroid disease, we found that higher FT4 and lower TSH levels were associated with better pulmonary function in the univariate analysis, but these associations were not



**Table 4. Age-specific association between thyroid function and pulmonary function.**

| Pulmonary function | FT4 | | TSH | | TPOAb | |
|---|---|---|---|---|---|---|
| | β (95% CI) | *P*-value | β (95% CI) | *P*-value | β (95% CI) | *P*-value |
| 40-59 years (n = 1,822) | | | | | | |
| FVC | | | | | | |
| Crude | 1.119 (0.720, 1.518) | <0.001 | −0.060 (−0.085, −0.035) | <0.001 | −0.980 (−1.580, −0.380) | 0.002 |
| Adjusted | 0.096 (−0.129, 0.321) | 0.413 | −0.013 (−0.021, −0.005) | <0.001 | −0.320 (−0.905, 0.265) | 0.289 |
| FEV1 | | | | | | |
| Crude | 0.857 (0.420, 1.294) | <0.001 | −0.041 (−0.066, −0.016) | <0.001 | −0.750 (−1.360, −0.140) | 0.015 |
| Adjusted | 0.152 (−0.029, 0.333) | 0.099 | −0.010 (−0.022, 0.002) | 0.107 | −0.280 (−0.880, −0.320) | 0.024 |
| FEV1/FVC | | | | | | |
| Crude | −0.08 (−0.260, 0.100) | 0.378 | 0.002 (0.0004, 0.004) | 0.011 | 0.110 (0.015, 0.205) | 0.031 |
| Adjusted | 0.019 (−0.012, 0.050) | 0.103 | 0.000 (−0.001, 0.001) | 0.830 | 0.030 (−0.050, 0.110) | 0.464 |
| ≥60 years (n = 804) | | | | | | |
| FVC | | | | | | |
| Crude | 0.274 (−0.124, 0.672) | 0.177 | 0.006 (−0.009, 0.021) | 0.436 | −0.840 (−1.550, −0.130) | 0.020 |
| Adjusted | −0.398 (−0.749, −0.047) | 0.026 | 0.018 (−0.017, 0.053) | 0.305 | −0.320 (−1.060, 0.420) | 0.390 |
| FEV1 | | | | | | |
| Crude | −0.025 (−0.320, 0.270) | 0.869 | 0.004 (−0.011, 0.019) | 0.602 | −0.710 (−1.300, −0.120) | 0.019 |
| Adjusted | −0.457 (−0.740, −0.174) | 0.002 | 0.004 (−0.018, 0.026) | 0.742 | −0.270 (−0.950, 0.410) | 0.429 |
| FEV1/FVC | | | | | | |
| Crude | −0.066 (−0.100, −0.032) | <0.001 | 0.000 (−0.002, 0.002) | 0.782 | 0.060 (−0.020, 0.140) | 0.138 |
| Adjusted | −0.048 (−0.091, −0.005) | 0.030 | −0.002 (−0.008, 0.004) | 0.389 | 0.010 (−0.080, 0.100) | 0.822 |

Adjusted for age, sex, body mass index, smoking, drinking, comorbidity and urine iodine.

Abbreviations fT4, free thyroxine; TSH, thyroid-stimulating hormone; TPOAb, thyroid peroxidase antibodies; forced expiratory volume in 1 second; FVC, forced vital capacity.

significant after adjusting for lifestyle and demographic factors. In contrast, TPOAb levels remained significantly associated with reduced FEV1, even after full adjustment, suggesting a role for thyroid autoimmunity in subclinical pulmonary impairment. Stratified analyses showed that the impact of thyroid function on pulmonary outcomes may differ depending on individual characteristics such as age or BMI. FT4 were inversely associated with pulmonary function in those aged ≥60 years, TSH was negatively associated with FEV1 in overweight individuals, and TPOAb levels were linked to reduced FVC and FEV1 in participants with a lower BMI (≤23 kg/m²). These findings highlight the importance of considering age and body composition when assessing the pulmonary effects on thyroid function and autoimmunity.

Previous studies on the relationship between thyroid function and pulmonary function have primarily focused on patients with overt thyroid disorders or chronic respiratory diseases [3,4]. More recent population-based studies have



**Table 5. An association between thyroid function and pulmonary function according to BMI.**

| Pulmonary function | FT4 | | TSH | | TPOAb | |
|---|---|---|---|---|---|---|
| | β (95% CI) | *P*-value | β (95% CI) | *P*-value | β (95% CI) | *P*-value |
| BMI ≤ 23 kg/m² (n = 951) | | | | | | |
| FVC | | | | | | |
| Crude | 1.159 (0.682, 1.636) | <0.001 | −0.030 (−0.060, 0.00) | 0.045 | −0.920 (−1.580, −0.260) | 0.006 |
| Adjusted | 0.120 (−0.197, 0.437) | 0.460 | 0.004 (−0.017, 0.025) | 0.677 | −0.310 (−1.342, −0.298) | 0.019 |
| FEV1 | | | | | | |
| Crude | 0.803 (0.312, 1.294) | <0.001 | −0.015 (−0.051, 0.021) | 0.243 | −0.730 (−1.320, −0.140) | 0.014 |
| Adjusted | 0.026 (−0.202, 0.254) | 0.848 | 0.009 (−0.007, 0.025) | 0.250 | −0.280 (−0.950, −0.183) | 0.047 |
| FEV1/FVC | | | | | | |
| Crude | −0.028 (−0.063, 0.007) | 0.105 | 0.002 (0.000, 0.004) | 0.020 | 0.095 (0.010, 0.180) | 0.031 |
| Adjusted | −0.018 (−0.051, 0.015) | 0.277 | 0.002 (−0.001, 0.005) | 0.129 | 0.020 (−0.070, 0.110) | 0.658 |
| BMI > 23 kg/m² (n = 1 675) | | | | | | |
| FVC | | | | | | |
| Crude | 0.823 (0.521, 1.125) | <0.001 | −0.060 (−0.082, −0.038) | <0.001 | −0.670 (−1.210, −0.130) | 0.015 |
| Adjusted | −0.187 (−0.446, 0.072) | 0.099 | −0.011 (−0.036, 0.014) | 0.243 | −0.220 (−0.890, 0.450) | 0.508 |
| FEV1 | | | | | | |
| Crude | 0.592 (0.267, 0.917) | <0.001 | −0.040 (−0.066, −0.014) | <0.001 | −0.600 (−1.110, −0.090) | 0.022 |
| Adjusted | −0.119 (−0.370, 0.132) | 0.183 | −0.015 (−0.029, −0.001) | 0.039 | −0.240 (−0.910, 0.430) | 0.489 |
| FEV1/FVC | | | | | | |
| Crude | −0.018 (−0.045, 0.009) | 0.115 | 0.002 (0.000, 0.004) | 0.018 | 0.080 (0.002, 0.158) | 0.042 |
| Adjusted | 0.002 (−0.033, 0.037) | 0.886 | −0.001 (−0.004, 0.002) | 0.228 | 0.010 (−0.085, 0.105) | 0.834 |

Adjusted for age, sex, body mass index, smoking, drinking, comorbidity and urine iodine.

Abbreviations fT4, free thyroxine; TSH, thyroid-stimulating hormone; TPOAb, thyroid peroxidase antibodies; forced expiratory volume in 1 second; FVC, forced vital capacity.

expanded this scope by focusing on euthyroid individuals. A study using NHANES data showed that reduced sensitivity to thyroid hormones, assessed using the Thyroid Feedback Quantile-based Index (TFQI) and the FT3/FT4 ratio, was significantly associated with impaired pulmonary function even within the euthyroid range [17]. Similarly, a Korean nationwide cohort study demonstrated that higher FT4 levels were associated with an increased risk of obstructive spirometry patterns in euthyroid individuals aged 45–65 years [8].

A major strength of our study is in the identification of thyroid autoimmunity, particularly elevated TPOAb levels, as an independent factor associated with impaired pulmonary function. In addition to their diagnostic utility in autoimmune thyroiditis, elevated TPOAb levels have been linked to metabolic alterations in Hashimoto's thyroiditis and autoimmune development in type 2 diabetes with metabolic dysfunction-associated steatotic liver disease, highlighting their broader influence on metabolic and immune pathways [18,19]. Higher TPOAb titers were also independently associated with

elevated high-sensitivity C-reactive protein levels in non-obese euthyroid individuals, indicating the presence of a low-grade inflammatory state [20]. This persistent inflammatory state may underlie subtle pulmonary dysfunction, in line with mechanisms proposed in chronic respiratory conditions, and interstitial pneumonia [7,21,22]. TPOAb remained significantly associated with reduced FEV1, whereas its association with FVC was attenuated. This suggests that thyroid autoimmunity may predominantly affect airway dynamics rather than lung volumes. Prior studies showed that subtle imbalances following TPOAb-induced dysfunction may primarily affect airway function, leading to functional impairments in the absence of reduced lung volumes [17].

Lifestyle factors such as smoking, obesity, alcohol intake, physical activity, and iodine consumption are known to influence both thyroid and pulmonary function. In our study, adjusting for these variables attenuated the association of FT4 and TSH levels with pulmonary function indices. Smoking is associated with suppressed TSH and elevated FT4 levels, possibly via sympathetic stimulation or altered deiodinase activity [23,24]. Physical activity is positively associated with pulmonary capacity, and has been shown to modulate thyroid hormone levels and reduce TPOAb titers. A recent longitudinal study of more than 20,000 participants from the U.K. Biobank reported that sustained low physical activity was associated with accelerated declines in FEV1 and FVC [25]. A Korean cross-sectional study found that moderate physical activity was associated with favorable thyroid hormone profiles, including lower TSH and TPOAb levels [16] Future research may benefit from structural equation modeling to disentangle the interrelated effects of lifestyle, thyroid function, and pulmonary outcomes [26].

The subgroup analyses revealed that the association between thyroid function and pulmonary outcomes varies according to sex, age, and body composition. In contrast to the sex-specific associations observed in men, the absence of significant associations in women may reflect differences in immune regulation or autoantibody expression between the sexes. Estrogen has been shown to modulate both thyroid autoimmunity and pulmonary inflammation, potentially attenuating the systemic impact of TPOAb in women through ERβ-mediated immunosuppressive pathways [27]. In addition to sex differences, aging may also modify thyroid–lung interactions. Age-related changes in thyroid hormone receptor sensitivity have been proposed in previous studies as a potential explanation for inverse associations observed in older adults, particularly in the context of altered tissue responsiveness despite normal hormone levels [28]. While a recent meta-analysis reported a J-shaped association between TSH levels and frailty in older adults, our findings did not reveal a significant association between TSH and pulmonary function, suggesting that TSH may play a more limited role in lung function decline than in frailty-related processes [29]. Body composition also appeared to influence thyroid–lung interactions, with the inverse association between TPOAb and pulmonary indices most evident among individuals with a BMI ≤ 23 kg/m². Previous studies have shown that individuals with a lower BMI tend to have smaller baseline lung volumes and may experience more pronounced declines in pulmonary function over time, even with modest weight gain or stable weight [30]. This physiological constraint may increase their vulnerability to autoimmune-related impairments. This offers a plausible explanation for the BMI-specific findings in our study.

Our study had several limitations. First, serum T3 levels were not available in the KNHANES dataset, which limited our ability to evaluate the association between pulmonary function and the biologically active form of thyroid hormones. Second, the cross-sectional design of this study prevented us from drawing conclusions regarding causality. Nonetheless, we attempted to provide plausible physiological explanations for the observed associations based on the existing literature. Although we observed biologically plausible associations and referenced supporting evidence from the literature, longitudinal studies are required to confirm temporal relationships. Third, our analysis was restricted to spirometric parameters (FEV1, FVC, and FEV1/FVC ratio). Therefore, comprehensive assessments such as total lung capacity, diffusing capacity, and peak expiratory flow rate were not available [31–33]. However, the spirometric indices employed were sufficient to identify both obstructive and restrictive ventilatory patterns and to perform subgroup analyses in participants with COPD. Fourth, potential residual confounding factors cannot be completely ruled out. Although we adjusted for multiple lifestyle factors including smoking, drinking, BMI, and physical activity, other environmental exposures such as air pollutants,



endocrine-disrupting chemicals, and dietary micronutrients may have influenced both thyroid function and pulmonary outcomes [34,35]. Despite these limitations, our study has several notable strengths. To our knowledge, this is one of the largest population-based studies to assess the link between thyroid function, including autoimmunity, and lung function in euthyroid individuals. Using nationally representative data enhances generalizability, and stratified analyses by age and BMI offer new insights into how thyroid–lung relationships may vary across subgroups.

## Conclusion

our study showed that even without overt thyroid dysfunction, thyroid hormone levels and thyroid autoimmunity are linked to subtle declines in lung function. These associations were more pronounced in older adults and individuals with lower BMI, indicating increased vulnerability in these groups. These findings suggest that thyroid autoimmunity may be a useful marker for identifying individuals at a risk of pulmonary decline. Future longitudinal studies are needed to clarify the biological mechanisms linking thyroid autoimmunity to reduced pulmonary function and to assess whether early detection of thyroid autoimmunity can help guide interventions to prevent pulmonary decline.

## Acknowledgments

This work was supported by the Research Institute of Medical Science, Eunpyeong St. Mary's Hospital, The Catholic University of Korea.

## Author contributions

**Conceptualization:** Jeongmin Lee.

**Data curation:** Kyung-Hun Sung, Jung-Min Lee, Jeongmin Lee.

**Investigation:** Mee Kyung Kim, Dong-Jun Lim, Sang-Ah Chang, Jeongmin Lee.

**Methodology:** Kyung-Hun Sung, Jeongeun Kwak, Mee Kyung Kim, Dong-Jun Lim, Jung-Min Lee, Jeongmin Lee.

**Project administration:** Jeongmin Lee.

**Supervision:** Sang-Ah Chang, Jeongmin Lee.

**Validation:** Jeongeun Kwak, Mee Kyung Kim, Dong-Jun Lim, Jung-Min Lee.

**Visualization:** Kyung-Hun Sung.

**Writing – original draft:** Kyung-Hun Sung, Jeongmin Lee.

**Writing – review & editing:** Jeongmin Lee.

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
