## [Decision Letter · Decision Letter 0]

23 Sep 2025

Dear Dr. Lee,

Thank you for submitting your manuscript to PLOS ONE. After careful consideration, we feel that it has merit but does not fully meet PLOS ONE’s publication criteria as it currently stands. Therefore, we invite you to submit a revised version of the manuscript that addresses the points raised during the review process.

**Please modify the article in the light of comments of the reviewers. **plosone@plos.org . A rebuttal letter that responds to each point raised by the academic editor and reviewer(s). You should upload this letter as a separate file labeled 'Response to Reviewers'.A marked-up copy of your manuscript that highlights changes made to the original version. You should upload this as a separate file labeled 'Revised Manuscript with Track Changes'.An unmarked version of your revised paper without tracked changes. You should upload this as a separate file labeled 'Manuscript'.

We look forward to receiving your revised manuscript.

Kind regards,

Aamir Ijaz, MD, FCPS, FRCP, MCPS-HPE

Academic Editor

PLOS ONE

Journal Requirements:

2. In this instance it seems there may be acceptable restrictions in place that prevent the public sharing of your minimal data. However, in line with our goal of ensuring long-term data availability to all interested researchers, PLOS’ Data Policy states that authors cannot be the sole named individuals responsible for ensuring data access (http://journals.plos.org/plosone/s/data-availability#loc-acceptable-data-sharing-methods ).

Reviewers' comments:

Reviewer's Responses to Questions

**Comments to the Author**

1. Is the manuscript technically sound, and do the data support the conclusions?

Reviewer #1: Yes

Reviewer #2: Yes

2. Has the statistical analysis been performed appropriately and rigorously?

Reviewer #1: Yes

Reviewer #2: Yes

3. Have the authors made all data underlying the findings in their manuscript fully available?

Reviewer #1: Yes

Reviewer #2: Yes

4. Is the manuscript presented in an intelligible fashion and written in standard English?

Reviewer #1: Yes

Reviewer #2: Yes

Reviewer #1: 1. The authors should correct the references. The References should be in Journal Format.

2. The authors have mentioned that this study complied with the ethical standards

of the Declaration of Helsinki and was approved by the Catholic University of Korea,

Catholic Medical Center, Eunpyeong St. Mary’s Hospital Institutional Review Board

(approval no. PC25ZISI0065). Mention the Intuitional Ethical Clearance date also.

Reviewer #2: This study explored the relationship between thyroid function and lung performance in euthyroid Korean adults. Using data from over 2,600 participants aged 40 and above, the researchers measured lung capacity and airflow with spirometry tests. They analyzed thyroid hormone levels and the presence of thyroid peroxidase antibodies (TPOAb), adjusting for lifestyle and demographic factors. The results showed that higher free thyroxine correlated with better lung function, while higher TSH related to poorer outcomes. Importantly, elevated TPOAb levels were linked to reduced lung function, independent of hormone levels. These associations were stronger in older adults and those with lower body mass index. The findings suggest that even without overt thyroid disease, thyroid autoimmunity can impair pulmonary health. The study addressed an underexplored correlation between thyroid autoimmunity and lung function. While the findings are not yet clinically applicable, they provide valuable insight and contribute to the literature, highlighting the need for further research to clarify and establish potential clinical relevance.

**Do you want your identity to be public for this peer review?** For information about this choice, including consent withdrawal, please see our Privacy Policy

Reviewer #1: No

Reviewer #2: **Yes: ** Jawaher Alsughayyir

---

## [Author Response · Author response to Decision Letter 1]

21 Oct 2025

Response to reviewers

Reviewer 1

Comment 1: The authors should correct the references. The References should be in Journal Format.

Response to comment 1: All references have been thoroughly revised and reformatted according to the journal’s reference style requirements.

Comment 2. The authors have mentioned that this study complied with the ethical standards of the Declaration of Helsinki and was approved by the Catholic University of Korea,

Catholic Medical Center, Eunpyeong St. Mary’s Hospital Institutional Review Board

(approval no. PC25ZISI0065). Mention the Intuitional Ethical Clearance date also.

Response to comment 2.: We have now included the date of institutional ethical approval in the revised manuscript. The sentence has been updated as follow:

Eunpyeong St. Mary’s Hospital Institutional Review Board (approval no. PC25ZISI0065, April 10th 2025).

Reviewer 2. This study explored the relationship between thyroid function and lung performance in euthyroid Korean adults. Using data from over 2,600 participants aged 40 and above, the researchers measured lung capacity and airflow with spirometry tests. They analyzed thyroid hormone levels and the presence of thyroid peroxidase antibodies (TPOAb), adjusting for lifestyle and demographic factors. The results showed that higher free thyroxine correlated with better lung function, while higher TSH related to poorer outcomes. Importantly, elevated TPOAb levels were linked to reduced lung function, independent of hormone levels. These associations were stronger in older adults and those with lower body mass index. The findings suggest that even without overt thyroid disease, thyroid autoimmunity can impair pulmonary health. The study addressed an underexplored correlation between thyroid autoimmunity and lung function. While the findings are not yet clinically applicable, they provide valuable insight and contribute to the literature, highlighting the need for further research to clarify and establish potential clinical relevance.

Response to comment: We sincerely thank the reviewer for the insightful feedback.

---

## [Editor Report · Decision Letter 1]

2 Nov 2025

Role of thyroid autoimmunity independent of thyroid hormone levels in pulmonary function decline: A population-based study in euthyroid adults

PONE-D-25-38118R1

Dear Dr. Lee,

We’re pleased to inform you that your manuscript has been judged scientifically suitable for publication and will be formally accepted for publication once it meets all outstanding technical requirements.

Kind regards,

Aamir Ijaz, MD, FCPS, FRCP, MCPS-HPE

Academic Editor

PLOS ONE
---

## [Editor Report · Acceptance letter]

PONE-D-25-38118R1

PLOS ONE

Dear Dr. Lee,

I'm pleased to inform you that your manuscript has been deemed suitable for publication in PLOS ONE. Congratulations! Your manuscript is now being handed over to our production team.

Kind regards,

on behalf of

Professor Aamir Ijaz

Academic Editor

PLOS ONE